# Synthesis and Evaluation of NF-κB Inhibitory Activity of Mollugin Derivatives

**DOI:** 10.3390/molecules27227925

**Published:** 2022-11-16

**Authors:** Lin-Hao Zhang, Ming-Yue Li, Da-Yuan Wang, Xue-Jun Jin, Fen-Er Chen, Hu-Ri Piao

**Affiliations:** 1Key Laboratory of Natural Medicines of the Changbai Mountain, Ministry of Education, College of Pharmacy, Yanbian University, Yanji 133000, China; 2Engineering Center of Catalysis and Synthesis for Chiral Molecules, Department of Chemistry, Fudan University, Shanghai 200433, China

**Keywords:** mollugin, NF-κB inhibitor, anti-inflammatory activity, cytotoxicity, ADMET

## Abstract

(1) Background: Nuclear factor κB (NF-κB) is an important transcriptional regulator that regulates the inflammatory pathway and plays a key role in cellular inflammatory and immune responses. The presence of a high concentration of NF-κB is positively correlated with the severity of inflammation. Therefore, the inhibition of this pathway is an important therapeutic target for the treatment of various types of inflammation; (2) Methods: we designed and synthesized 23 mollugin derivatives and evaluated their inhibitory activity against NF-κB transcription; (3) Results: Compound **6d** exhibited the most promising inhibitory activity (IC_50_ = 3.81 µM) and did not show any significant cytotoxicity against the tested cell lines. Investigation of the mechanism of action indicated that **6d** down-regulated NF-κB expression, possibly by suppressing TNF-α-induced expression of the p65 protein. Most of the compounds exhibited potent anti-inflammatory activity. Compound **4f** was the most potent compound with 83.08% inhibition of inflammation after intraperitoneal administration, which was more potent than mollugin and the reference drugs (ibuprofen and mesalazine). ADMET prediction analysis indicated that compounds **6d** and **4f** had good pharmacokinetics and drug-like behavior; (4) Conclusions: Several series of mollugin derivatives were designed, synthesized, and evaluated for NF-κB inhibitory activity and toxicity. These results provide an initial basis for the development of 4f and 6d as potential anti-inflammatory agents.

## 1. Introduction

Inflammation is a common and important basic pathological process, but excessive inflammation can cause a physiological imbalance in tissues and organs [1], which becomes a serious threat to human health in allergic reactions, autoimmune diseases, and organ transplant rejection [2]. Currently, nonsteroidal anti-inflammatory drugs, such as indomethacin and ibuprofen, are used to treat acute inflammation, but their long-term use can cause adverse reactions, including bone tissue damage, gastrointestinal injury, liver damage, and toxic kidney injury [3]. Therefore, there is a need to develop new targeted compounds for the prevention and treatment of inflammation [4].

The nuclear factor κB (NF-κB) family of transcription factors plays a central role in coordinating the expression of genes that control inflammation, immune responses, cell proliferation, and a variety of other biological processes [5]. The NF-κB pathway has long been considered a prototypical pro-inflammatory signaling pathway, largely based on the role of NF-κB in the expression of pro-inflammatory genes, including for cytokines, chemokines, and adhesion molecules [6]. Activated NF-κB is transferred to the cell nucleus and when combined with promoters for pro-inflammatory genes, leads to enhanced gene expression and an increased inflammatory response, and ultimately tissue inflammatory damage [7]. Therefore, inhibition of NF-κB transcription activity may be an effective way to develop a lead compound for the treatment of inflammatory diseases [8].

Mollugin is a naphthoquinone compound isolated from the Rubiaceae plant [9]. Mollugin has many biological activities and is used for the treatment of cough, inflammation of the joints, uterine hemorrhage, and uteritis [10]. Mollugin has been shown to exhibit anti-inflammatory and anti-tumor effects via inhibition of the NF-κB signaling pathway [10,11,12,13]. Recently, various structural modifications of the mollugin core have been reported that improved the bioactive properties to generate potent anti-inflammatory agents [10,14]. Jiang et al. [15] reported that the conjugated structure of the naphthalene and pyran rings of mollugin are critical for physiological activity. Studies by Han et al. [16] indicated that replacement of the ester bond with an amide made the derivatives easier to salify, and not only improved the solubility, but significantly improved the inhibitory activity, compared with mesalazine. These compounds could be effectively used for the prevention and treatment of inflammatory bowel disease (Figure 1A,B). In addition, simultaneously extending the carbon chain and introducing different heterocyclic rings at C-28 also significantly improved the anti-inflammatory activity of mollugin (Figure 1C,D,E). However, there have been few reports on modification of the phenolic hydroxyl group of mollugin to improve the biological activity. In the present study, we designed and synthesized three series of 23 previously unreported mollugin derivatives modified at the C-6 phenolic hydroxyl group. The synthesized compounds were evaluated for transcriptional inhibitory activity against NF-κB and anti-inflammatory activity using the xylene-induced ear edema test in mice.

## 2. Results and Discussion

### 2.1. Chemistry

The synthetic pathway for the target compounds **4a–i**, **6a–k**, and **8a–c** is presented in Figure 1. Mollugin was prepared by the reaction of methyl 1,4-dihydroxy-2-naphthoate with 3-chloro-3-methyl-1-butyne followed by substitution and cyclization in anhydrous toluene. Compounds **4a–i** and **6a–k** were synthesized by the substitution reaction of mollugin with various N-substituted phenyl chloroacetamides or N-substituted aromatic heterocyclic chloroacetamides (except for **6f**). Compounds **8a–c** were synthesized by the substitution of the C-6 hydroxy group with N-aliphatic or N-phenyl bromoethylamine derivatives. All the derivatives were characterized by ^1^H and ^13^C NMR spectroscopy and high-resolution mass spectrometry.

### 2.2. Evaluation of Biological Activities

#### 2.2.1. Luciferase Reporter and MTT Analysis

The in vitro activities of the mollugin derivatives are summarized in Table 1, Table 2 and Table 3. To examine the effect of the mollugin derivatives on hypoxia–induced NF-κB transcriptional activity, the cytotoxicity of the derivatives against HeLa cells was evaluated using the MTT assay. Mollugin was used as a positive control.

The majority of the synthesized compounds inhibited NF-κB transcriptional activity, and 10 compounds had superior activity to that of mollugin. In series **4a–i**, compounds **4d** (IC_50_ = 88.25 µM), **4e** (IC_50_ = 55.12 µM), **4i** (IC_50_ = 22.93 µM), and **4f** (IC_50_ = 18.53 µM) showed increased inhibitory activity compared with the parent compound (mollugin IC_50_ > 100 µM) with no significant cell toxicity. In this series, it was evident that the effect of the substituents on the benzene ring was in the order para position > meta and ortho position, and no significant differences were observed between electron-withdrawing and electron-donating groups. In series **6a–k**, compound **6d** was the most potent with an IC_50_ value of 3.81 µM, but no clear structure–activity relationship could be found. Compounds **8a–c**, which did not contain any amide moieties, displayed no activity or toxicity. Compounds in series **6a–k** exhibited better activity compared with those in **4a–i**, indicating that an N-substituted aromatic heterocycle moiety was more beneficial than an N-substituted phenyl moiety. Comparing the inhibitory activities of the compounds in series **4** and **6** with those in series **8** indicated that the introduction of an amide bond could significantly improve the activity.

Of all the derivatives tested, compound **6d** exhibited the most significant inhibitory activity against NF-κB and was selected for further evaluation. As shown in Figure 2, compound **6d** exhibited dose-dependent inhibition of NF-κB luciferase activity in HeLa cells (Figure 2A) and did not adversely affect the cell viability at a concentration of 30 µM (Figure 2B).

#### 2.2.2. Western Blot Analysis

To understand the mechanism underlying the ability of **6d** to suppress NF-κB transcriptional activity, the effects of **6d** on TNF-α-induced phosphorylation of p65 were examined by Western blot analysis because the phosphorylation of p65 is vital to the NF-κB transcriptional activity [17]. Nuclear extracts were analyzed by Western blotting and the results are shown in Figure 3. Compound **6d** significantly inhibited TNF-α-induced p65 phosphorylation in a dose-dependent manner. Therefore, we speculated that compound 6d might inhibit NF-κB transcriptional activity by inhibiting the expression of p65.

#### 2.2.3. Anti-Inflammatory Activity

Next, some of the synthesized compounds were screened for anti-inflammatory activity using the xylene-induced ear edema test in mice. The potency of the tested compounds was determined relative to the ability to prevent edema. Dimethyl sulfoxide was used as the vehicle, and ibuprofen and mesalazine were used as reference drugs. As shown in Table 4, most of the compounds showed significant activity with inhibition in the range of 55.81% to 83.08%. Compounds in series **4a–i** and **6a–k** were more active than the compounds in series **8a–c**; in particular, compounds **4f** and **6d** were the most potent with inhibition rates of 83.08% and 76.77%, respectively, and were more potent than mollugin and the positive control drugs (ibuprofen and mesalazine).

Compounds **4f** and **6d**, which had the best anti-inflammatory activity of the designed compounds, were selected for further evaluation. A dose of 100 mg/kg was orally administered at different intervals (1, 2, 3, 4, 5, and 24 h) after xylene application. As shown in Table 5, the activity of compounds **4f** and **6d** showed a regular increase as the time interval increased until a peak was reached at 6 h (77.78% and 67.68%, respectively), which was more potent activity than that of ibuprofen (47.47%). The activity of compounds **4f** and **6d** was also screened at concentrations of 100, 50, and 25 mg/kg at 6 h after oral administration. As shown in Table 6, compounds **4f** and **6d** showed the highest anti-inflammatory activity at 100 mg/kg with 74.32% and 62.39% inhibition, respectively.

### 2.3. Prediction of ADMET Properties

To investigate the druggability of the most active compounds **4f** and **6d**, we conducted an ADMET prediction experiment [18]. The ADMET plot of compounds **4f** and **6d** is shown in Figure 4 and Table 6. The predictions for compounds **4f** and **6d** were within the ellipses of the 95% and 99% human intestinal absorption (HIA) confidence regions, indicating that they could be well absorbed (Figure 4). Compound **6d** lay between the 95% and 99% oval-shaped confidence regions of the blood brain barrier (BBB), and compound **4f** lay in the 95% oval-shaped confidence regions of the BBB, suggesting that compound **6d** might have better BBB penetration [19]. The parameter values, other than the two predicted parameters HIA and BBB (AS, CYP2D6, HT, and PPB) are shown in Table 7. Compounds **4f** and **6d** showed very low aqueous solubility, which will limit the potential for oral administration. The two compounds had plasma protein-binding capacities similar to those of the positive control mollugin, but neither were CYP2D6 inhibitors, suggesting weak drug–drug interactions during metabolism. As the ADMET predicted values of the two compounds were within the allowable range for human use, these compounds might have good pharmacokinetic and pharmacodynamic properties.

## 3. Materials and Methods

### 3.1. Experimental Compounds

#### 3.1.1. General Procedures

All chemicals and spectral grade solvents were obtained commercially and were used without further purification. Solvents were dried and used according to standard procedures. All chemical reactions were monitored by thin-layer chromatography (TLC). The melting points of all compounds were determined by capillary method (temperature uncorrected). The ^1^HNMR and ^13^CNMR spectra were determined on a BRUKER AV-300 (Bruker Daltonics Inc, USA) using Chloroform-d (CDCL_3_) as the solvent and the chemical shift unit is ppm. High resolution mass spectra of the compounds were determined by UHPLC-Q-Obritrap MS (Thermo Scientific, Waltham, MA, USA).

#### 3.1.2. Procedure for the Synthesis of the Mollugin 2

Methyl 1,4-dihydroxynaphthalene-2-carboxylate (1) was dissolved in toluene solution, and 3 equivalents of 3-Chloro-3-methyl-1-butyne, 2 equivalents K_2_CO_3_, 1 equivalent CuCl were added portion wise. It was heated to reflux with N_2_ atmosphere for 24 h. The reaction process was monitored by TLC. The mixture was filtered and concentrated to give crude compound. The crude product was purified by column chromatography (petroleum ether: ethyl acetate = 10:1) to obtain Mollugin **2**.

#### 3.1.3. General Procedure for the Synthesis of the Intermediate **3a–i**

To a mixture of N-substituted phenylamine in acetone was added chloroacetyl chloride dropwise at 0 °C temperature. After stirring for 30 min, it was heated to reflux for 6 h. The mixture was quenched with ice water. After separation, the aqueous phase was extracted with dichloromethane, dried over with anhydrous sodium sulfate, filtered, and concentrated to give the crude compound. The crude compound was purified by column chromatography to obtain intermediate **3a–i**.

#### 3.1.4. General Procedure for the Synthesis of the Intermediate **5a–k**

To a mixture of N-substituted aromatic heterocycle amine in acetone was added chloroacetyl chloride dropwise at 0 °C temperature. After stirring for 30 min, it was heated to reflux for 24 h. The mixture was quenched with ice water. After separation, the aqueous phase was extracted with ethyl acetate, dried over with anhydrous sodium sulfate, filtered, and concentrated to give the crude compound. The crude compound was purified by column chromatography to obtain intermediate **5a–k**.

#### 3.1.5. General Procedure for the Synthesis of the Intermediate **7a–b**

A mixture of N-substituted phenylamine, equimolar K_2_CO_3_, catalytic amount KI, and more than 2 equivalents 1,2-dibromoethane in DMF was stirred at 50 °C for 17 h. The reaction was monitored by TLC. After quenching the reaction with cold water, the reaction mixture was filtered to get the crude product, which was purified by column chromatography to give pure intermediate **7a–b**.

#### 3.1.6. General Procedure for the Preparation of Target Compounds **4a–i**

To a mixture of Mollugin **2** in acetone was added K_2_CO_3_, KI and various substituted phemyl chroroacetamide **3a–i** in turn. The reaction was stirred for 12 h under refluxing temperatures. The mixture was quenched with water. After separation, the aqueous phase was extracted with ethyl acetate, dried over with anhydrous sodium sulfate, filtered, and concentrated to give the crude compound. The crude compound was purified by column chromatography (petroleum ether: ethyl acetate = 10:1) to obtain the desired products **4a–i**. The structural formula and yield of compound **4a–i** are shown in Table 8 and further characterized by the physical and spectroscopic data shown below.

##### Methyl 2,2-dimethyl-6-(2-oxo-2-(phenylamino)ethoxy)-2H-benzo [h] chromene-5-carboxyate (**4a**)

The spectrum of ^1^H NMR, ^13^C NMR and HRMS of compounds **4a** are shown in Appendix A.

M.p.118–120 °C. ^1^H-NMR (300 MHz, CDCl_3_): δ 1.54 (s, 6H), 3.92 (s, 3H), 4.7 (s, 2H), 5.73 (d, 1H, J = 6.0 Hz), 6.46 (d, 1H, J = 6.0 Hz), 7.18 (t, 1H, J = 3.0 Hz), 7.4 (t, 2H, J = 3.0 Hz), 7.54–7.56 (m, 2H), 7.71 (d, 2H, J = 3.0 Hz), 7.94–7.96 (m, 1H), 8.24–8.26 (m, 1H), 8.77 (s, 1H). ^13^C-NMR (125 MHz, CDCl_3_): δ 27.86, 29.84, 52.83, 58.14, 74.54, 112.61, 119.84, 119.98, 120.56, 122.06, 122.93, 124.83, 127.00, 127.47, 127.50, 127.67, 129.30, 130.64, 137.40, 145.04, 146.08, 166.56, 167.51. HRMS (ESI) *m*/*z* calcd for C_25_H_23_N0_5_^+^ (M + H^+^) 418.1640, found 418.1649.

##### Methyl 6-(2-((4-chlorophenyl)amino)-2-oxoethoxy)-2,2-dimethyl-2H-benzo [h] chromene-5-carboxylate (**4b**)

The spectrum of ^1^H NMR, ^13^C NMR and HRMS of compounds **4b** are shown in Appendix A.

M.p.122–124 °C. ^1^H-NMR (300 MHz, CDCl_3_): δ 1.54 (s, 6H), 3.91 (s, 3H), 4.69 (s, 2H), 5.73 (d, 1H, J = 6.0 Hz), 6.46 (d, 1H, J = 6.0 Hz), 7.37 (d, 2H, J = 6.0 Hz), 7.55 (t, 2H, J = 3.0 Hz), 7.67–7.69 (d, 2H, J = 6.0 Hz), 7.93–7.95 (m, 1H), 8.24–8.26 (m, 1H), 8.82 (s, 1H). ^13^C-NMR (125 MHz, CDCl_3_): δ 27.86, 29.84, 52.82, 65.71, 74.53, 112.60, 119.84, 120.43, 121.19, 122.03, 122.97, 127.04, 127.47, 127.53, 127.71, 128.99, 129.32, 129.81, 130.66, 131.05, 136.04, 145.10, 146.14, 166.63, 167.56. HRMS (ESI) *m*/*z* calcd for C_25_H_22_ClN0_5_^+^ (M + H^+^) 452.1249, found 452.1259.

##### Methyl 6-(2-((4-bromophenyl)amino)-2-oxoethoxy)-2,2-dimethyl-2H-Benzo [h] chromene-5-carboxylate (**4c**)

The spectrum of ^1^H NMR, ^13^C NMR and HRMS of compounds **4c** are shown in Appendix A.

M.p.125–127 °C. ^1^H-NMR (300 MHz, CDCl_3_): δ 1.53 (s, 6H), 3.91 (s, 3H), 4.69 (s, 2H), 5.73 (d, 1H, J = 9.0 Hz), 6.46 (d, 1H, J = 9.0 Hz), 7.51–7.65 (m, 6H), 7.91–7.94 (m, 1H), 8.25 (t, 1H, J = 3.0 Hz), 8.84 (s, 1H). ^13^C-NMR (125 MHz, CDCl_3_): δ 27.43, 29.42, 52.40, 65.28, 74.12, 112.17, 116.99, 119.41, 119.98, 121.08, 121.59, 122.53, 126.60, 127.09, 127.27, 130.22, 131.84, 136.10, 144.67, 145.70, 166.21, 167.13. HRMS (ESI) *m*/*z* calcd for C_25_H_22_BrN0_5_^+^ (M + H^+^) 496.0747, found 496.0754.

##### Methyl 6-(2-((4-methoxyphenyl)amino)-2-oxoethoxy)-2,2-dimethyl-2H-benzo [h] chromene-5-carboxylate (**4d**)

The spectrum of ^1^H NMR, ^13^C NMR and HRMS of compounds **4d** are shown in Appendix A.

M.p.126–128 °C. ^1^H-NMR (300 MHz, CDCl_3_): δ 1.53 (s, 6H), 3.82 (s, 3H), 3.92 (s, 3H), 4.69 (s, 2H), 5.73 (d, 1H, J = 9.0 Hz), 6.46 (d, 1H, J = 9.0 Hz), 6.94 (d, 2H, J = 9.0Hz), 7.53–7.56 (m, 2H), 7.61–7.64 (d, 2H, J = 9.0 Hz), 7.93–7.97 (m, 1H), 8.23–8.26 (m, 1H), 8.68 (s, 1H). ^13^C-NMR (125 MHz, CDCl_3_): δ 27.86, 29.84, 52.83, 53.56, 65.71, 74.53, 112.60, 114.43, 119.83, 120.58, 121.66, 122.08, 122.92, 126.99, 127.45, 127.52, 127.64, 128.98, 130.63, 145.06, 146.05, 156.80, 166.30, 167.52. HRMS (ESI) *m*/*z* calcd for C_26_H_25_N0_6_^+^ (M + H^+^) 448.1751, found 448.1754.

##### Methyl 2,2-dimethyl-6-(2-oxo-2-(p-tolylamino)ethoxy)-2H-benzo [h] chromene-5-carboxylate (**4e**)

The spectrum of ^1^H NMR, ^13^C NMR and HRMS of compounds **4e** are shown in Appendix A.

M.p.120–122 °C. ^1^H-NMR (300 MHz, CDCl_3_): δ 1.54 (s, 6H), 2.35 (s, 3H), 3.92 (s, 3H), 4.69 (s, 2H), 5.73 (d, 1H, J = 9.0 Hz), 6.47 (d, 1H, J = 9.0 Hz), 7.21 (d, 2H, J = 9.0 Hz), 7.52–7.56 (m, 4H), 7.93–7.97 (m, 1H), 8.23–8.26 (m, 1H), 8.72 (s, 1H). ^13^C-NMR (125 MHz, CDCl_3_): δ 21.05, 27.84, 29.83, 52.82, 65.70, 74.53, 112.59, 119.82, 120.00, 120.59, 122.07, 122.90, 126.97, 127.44, 127.51, 127.63, 128.92, 129.77, 130.61, 131.06, 134.45, 134.84, 145.23, 146.02, 166.40, 167.58. HRMS (ESI) *m*/*z* calcd for C_26_H_25_N0_5_^+^ (M + H^+^) 432.1800, found 432.1805.

##### Methyl 6-(2-((4-fluorophenyl)amino)-2-oxoethoxy)-2,2-dimethyl-2H-benzo [h] chromene-5-carboxylate (**4f**)

The spectrum of ^1^H NMR, ^13^C NMR and HRMS of compounds **4f** are shown in Appendix A.

M.p.132–134 °C. ^1^H-NMR (300 MHz, CDCl_3_): δ 1.54 (s, 6H), 3.92 (s, 3H), 4.70 (s, 2H), 5.74 (d, 1H, J = 9.0 Hz), 6.46 (d, 1H, J = 9.0 Hz), 7.09 (t, 2H, J = 9.0 Hz), 7.53–7.57 (m, 2H), 7.67–7.71 (m, 2H), 7.92–7.96 (m, 1H), 8.23–8.27 (m, 1H), 8.79 (s, 1H). ^13^C-NMR (125 MHz, CDCl_3_): δ 27.86, 29.84, 52.82, 63.56, 74.51, 112.60, 115.81, 116.11, 119.84, 120.48, 121.64, 121.74, 122.04, 122.96, 127.03, 127.51, 127.69, 130.66, 133.50, 145.08, 146.12, 153.50, 161.19, 166.53, 167.56. HRMS (ESI) *m*/*z* calcd for C_25_H_22_FN0_5_^+^ (M + H^+^) 436.1549, found 436.1554.

##### Methyl 2,2-dimethyl-6-(2-oxo-2-(o-tolylamino)ethoxy)-2H-Benzo [h] chromene-5-carboxylate (**4g**)

The spectrum of ^1^H NMR, ^13^C NMR and HRMS of compounds **4g** are shown in Appendix A.

M.p.122–124 °C. ^1^H-NMR (300 MHz, CDCl_3_): δ 1.54 (s, 6H), 2.38 (s, 3H), 3.92 (s, 3H), 4.74(s, 2H), 5.71 (d, 1 H, J = 9.0 Hz), 6.44 (d, 1H, J = 9.0 Hz), 7.12 (t, 1H, J = 9.0 Hz), 7.54–7.57 (m, 2H), 7.97–8.00 (m, 1H), 8.12 (d, 1H, J = 9.0 Hz), 8.24–8.27 (m, 1H), 8.68 (s, 1H). ^13^C-NMR (125 MHz, CDCl_3_): δ 17.76, 27.84, 52.74, 53.42, 74.54, 112.63, 119.82, 120.69, 121.97, 122.14, 122.96, 125.27, 126.99, 127.09, 127.45, 127.50, 127.66, 128.43, 130.51, 130.57, 135.33, 144.86, 146.09, 154.19, 166.48, 167.32. HRMS (ESI) *m*/*z* calcd for C_26_H_25_N0_5_^+^ (M + H^+^) 432.1800, found 432.1805.

##### Methyl 2,2-dimethyl-6-(2-oxo-2-(m-tolylamino)ethoxy)-2H-benzo [h] chromene-5-carboxylate (**4h**)

The spectrum of ^1^H NMR, ^13^C NMR and HRMS of compounds **4h** are shown in Appendix A.

M.p.122–124 °C. ^1^H-NMR (300 MHz, CDCl_3_): δ 1.54 (s, 6H), 2.39 (s, 3H), 3.92 (s, 3H), 4.69 (s, 2H), 5.70 (d, 1H, J = 12.0 Hz), 6.44 (d, 1H, J = 9.0 Hz), 6.98 (d, 1H, J = 9.0 Hz), 7.28–7.30 (m, 1H), 7.49–7.58 (m, 4H), 7.93–7.96 (m, 1H), 8.23–8.26 (m, 1H), 8.71 (s, 1H). ^13^C-NMR (125 MHz, CDCl_3_): δ 21.66, 27.84, 50.98, 52.83, 74.52, 112.60, 117.08, 119.82, 120.60, 122.05, 122.91, 125.65, 126.98, 127.45, 127.65, 129.11, 130.63, 137.29, 139.24, 145.01, 146.06, 166.51, 167.48. HRMS (ESI) *m*/*z* calcd for C_26_H_25_N0_5_^+^ (M + H^+^) 432.1801, found 432.1805.

##### Methyl 6-(2-((4-(cyanomethyl)phenyl)amino)-2-oxoethoxy)-2,2-dimethyl-2H-benzo [h] chromene-5-carboxylate (**4i**)

The spectrum of ^1^H NMR, ^13^C NMR and HRMS of compounds **4i** are shown in Appendix A.

M.p.115–117 °C. ^1^H-NMR (300 MHz, CDCl_3_: δ 1.54 (s, 6H), 3.75 (s, 2H), 3.91 (s, 3H), 4.70 (s, 2H), 5.70 (d, 1H, J = 12.0 Hz), 6.43 (d, 1H, J = 9.0 Hz), 7.33 (d, 2H, J = 9.0 Hz), 7.54–7.56 (m, 2H), 7.73–7.76 (m, 2H), 7.92–7.95 (m, 1H), 8.23–8.26 (m, 1H), 8.86 (s, 1H). ^13^C-NMR (125 MHz, CDCl_3_): δ 23.29, 27.83, 52.82, 53.91, 74.52, 112.59, 117.95, 119.82, 120.49, 122.02, 122.95, 126.05, 127.01, 127.50, 127.68, 128.87, 130.65, 137.33, 145.07, 146.11, 166.72, 167.53. HRMS (ESI) *m*/*z* calcd for C_26_H_22_N_2_0_5_^+^ (M + H^+^) 457.1764, found 457.1758.

#### 3.1.7. General Procedure for the Preparation of Target Compounds **6a–k**

To a mixture of Mollugin **2** in acetone was added K_2_CO_3_, KI and N-substituted aromatic heterocycle chroroacetamides **5a–k** in turn. The reaction was stirred for 24 h under refluxing temperatures. The mixture was quenched with water. After separation, the aqueous phase was extracted with ethyl acetate, dried over with anhydrous sodium sulfate, filtered, and concentrated to give the crude compound. The crude compound was purified by column chromatography (petroleum ether: ethyl acetate = 5:1) to obtain the desired products **6a–k**. The structural formula and yield of compound **6a–k** are shown in Table 8 and further characterized by the physical and spectroscopic data shown below.

##### Methyl 6-(2-((1-benzylpiperidin-4-yl)amino)-2-oxoethoxy)-2,2-dimethyl-2H-benzo [h] chromene-5-carboxylate (**6a**)

The spectrum of ^1^H NMR, ^13^C NMR and HRMS of compounds **6a** are shown in Appendix A.

M.p.115–117 °C. ^1^H-NMR (300 MHz, CDCl_3_: δ 1.52 (s, 6H), 1.67–1.77 (m, 2H), 2.03 (d, 2H, J = 12.0Hz), 2.23–2.26 (m, 2H), 2.93 (s, 2H), 3.59 (s, 2H), 3.94 (s, 3H), 4.54 (s, 2H), 5.68 (d, 1H, J = 12.0 Hz), 6.41 (d, 1H, J = 9.0 Hz), 7.29–7.36 (m, 5H), 7.50–7.55 (m, 2H), 7.86–7.89 (m, 1H), 8.21–8.24 (m, 1H). ^13^C-NMR (125 MHz, CDCl_3_): δ 27.84, 29.41, 29.83, 31.87, 52.70, 53.94, 62.93, 69.63, 112.56, 119.83, 120.61, 122.03, 122.88, 126.94, 127.34, 127.48, 128.50, 129.48, 130.57, 144.99, 145.88, 167.36, 167.94. HRMS (ESI) *m*/*z* calcd for C_31_H_34_N_2_0_5_^+^ (M + H^+^) 515.2526, found 515.2540.

##### Methyl 2,2-dimethyl-6-(2-((3-methylpyridin-4-yl)amino)-2-oxoethoxy)-2H-benzo [h] chromene-5-carboxylate (**6b**)

The spectrum of ^1^H NMR, ^13^C NMR and HRMS of compounds **6b** are shown in Appendix A.

M.p.123–125 °C. ^1^H-NMR (300 MHz, CDCl_3_: δ 1.53 (s, 6H), 2.36 (s, 3H), 3.9 (s, 3H), 4.74 (s, 2H), 5.71 (d, 1H, J = 9.0 Hz), 6.44 (d, 1H, J = 9.0 Hz), 7.51–7.58 (m, 3H), 7.70–7.73 (m, 1H), 7.93–7.96 (m, 1H), 8.24–8.27 (m, 1H), 8.37–8.38 (m, 1H), 8.92 (s,1H). ^13^C-NMR (125 MHz, CDCl_3_): δ 14.36, 27.82, 52.72, 65.70, 74.67, 119.85, 120.36, 121.84, 123.04, 127.05, 127.43, 127.62, 127.81, 128.98, 130.70, 131.05, 132.46, 143.09, 144.94, 146.31, 148.77, 150.79, 167.16, 167.25. HRMS (ESI) *m*/*z* calcd for C_25_H_24_N_2_0_5_^+^ (M + H^+^) 433.1746, found 433.1758.

##### Methyl 6-(2-((4-chlorobenzyl)amino)-2-oxoethoxy)-2,2-dimethyl-2H-benzo [h] chromene-5-carboxylate (**6c**)

The spectrum of ^1^H NMR, ^13^C NMR and HRMS of compounds **6c** are shown in Appendix A.

M.p.130–132 °C. ^1^H-NMR (300 MHz, CDCl_3_: δ 1.51 (s, 6H), 3.82 (s, 3H), 4.56–4.62 (m, 4H), 5.67 (d, 1H, J = 9.0 Hz), 6.39(d, 1H, J = 9.0 Hz), 7.33–7.42 (m, 4H), 7.47–7.51 (m, 2H), 7.83 (d, 1H, J = 6.0 Hz), 8.20 (d, 1H, J = 4.0 Hz). ^13^C-NMR (125 MHz, CDCl_3_): δ 27.36, 29.37, 41.99, 52.07, 65.22, 73.87, 112.07, 119.32, 120.10, 121.47, 122.41, 126.44, 126.88, 127.02, 128.52, 128.93, 130.09, 133.07,136.29, 144.43, 145.44, 166.87, 168.16. HRMS (ESI) *m*/*z* calcd for C_26_H_24_ClN0_5_^+^ (M + H^+^) 466.14029, found 466.14158.

##### Methyl 2,2-dimethyl-6-(2-oxo-2-(quinolin-6-ylamino)ethoxy)-2H-benzo [h] Chromene-5-carboxylate (**6d**)

The spectrum of ^1^H NMR, ^13^C NMR and HRMS of compounds **6d** are shown in Appendix A.

M.p.135–137 °C. ^1^H-NMR (300 MHz, DMSO-d6): δ 1.48 (s, 6H), 3.90 (s, 3H), 4.74 (s, 2H), 5.88 (d, 1H, J = 9.0 Hz), 6.40 (d, 1H, J = 12.0 Hz), 7.48–7.52 (m, 1H), 7.62–7.65 (m, 2H), 7.94–8.03 (m, 2H), 8.13–8.16 (m, 1H), 8.26–8.35 (m, 2H), 8.50 (s, 1H), 8.82 (s, 1H). ^13^C-NMR (125 MHz, DMSO-d6): δ 27.20, 29.59, 52.67, 54.90, 68.49, 74.40, 76.64, 111.93, 115.85, 119.05, 120.68, 121.81, 121.90, 122.76, 123.74, 125.82, 127.22, 127.52, 128.24, 129.53, 131.22, 135.61, 136.36, 144.46, 144.89, 145.23, 149.26, 166.59, 166.78. HRMS (ESI) *m*/*z* calcd for C_28_H_24_N_2_0_5_^+^ (M + H^+^) 469.1755, found 469.1758.

##### Methyl 2,2-dimethyl-6-(2-oxo-2-(pyridin-4-ylamino)ethoxy)-2H-benzo [h] chromene-5-carboxylate (**6e**)

The spectrum of ^1^H NMR, ^13^C NMR and HRMS of compounds **6e** are shown in Appendix A.

M.p.126–128 °C. ^1^H-NMR (300 MHz, DMSO-d6): δ 1.48 (s, 6H), 3.87 (s, 3H), 4.70 (s, 2H), 5.87 (d, 1H, J = 12.0 Hz), 6.39 (d, 1H, J = 9.0 Hz), 7.64 (s, 2H), 7.72 (s, 2H), 8.16–8.22 (m, 2H), 8.48 (s, 1H), 10.49 (s, 1H). ^13^C-NMR (125 MHz, DMSO-d6): δ 27.20, 48.58, 76.65, 90.68, 107.12, 113.41, 113.69, 115.24, 117.63, 118.98, 119.03, 120.66, 121.89, 122.09, 122.66, 125.80, 127.15, 127.53, 131.21, 145.11, 150.42, 167.58. HRMS (ESI) *m*/*z* calcd for C_24_H_22_N_2_0_5_^+^ (M + H^+^) 419.1601, found 419.1601.

##### Methyl 6-(2-(cyclopropylamino)-2-oxoethoxy)-2,2-dimethyl-2H-benzo [h] chromene-5-carboxylate (**6f**)

The spectrum of ^1^H NMR, ^13^C NMR and HRMS of compounds **6f** are shown in Appendix A.

M.p.134–136 °C. ^1^H-NMR (300 MHz, CDCl3: δ 0.87–0.90 (m, 4H), 1.52 (s, 6H), 2.89–2.91 (m, 1H), 3.94 (s, 3H), 4.54 (s, 2H), 5.68 (d, 1H, J = 9.0 Hz), 6.40 (d, 1H, J = 9.0 Hz), 7.50–7.53 (m, 2H), 7.84–7.87 (m, 1H), 8.20–8.23 (m, 1H). ^13^C-NMR (125 MHz, CDCl_3_): δ 6.63, 22.35, 27.84, 52.62, 53.55, 74.38, 112.52, 119.77, 121.99, 122.87, 126.91, 127.34, 127.49, 130.58, 144.93, 145.88, 167.39, 169.97. HRMS (ESI) *m*/*z* calcd for C_22_H_23_N0_5_^+^ (M + H^+^) 386.1642, found 382.1649.

##### Methyl 6-(2-(benzhydrylamino)-2-oxoethoxy)-2,2-Dimethyl-2H-benzo [h] chromene-5-carboxylate (**6g**)

The spectrum of ^1^H NMR, ^13^C NMR and HRMS of compounds **6g** are shown in Appendix A.

M.p.127–129 °C. ^1^H-NMR (300 MHz, CDCl_3_: δ 1.40 (s, 6H), 3.61 (s, 3H), 4.55 (s, 2H), 4.70 (s, 2H), 5.57 (d, 1H, J = 9.0 Hz), 6.29–6.36 (m, 2H), 7.26–7.66 (m, 12H), 7.78 (d, 1H, J = 9.0 Hz), 8.10 (d, 1H, J = 6.0 Hz). ^13^C-NMR (125 MHz, CDCl_3_): δ 27.37, 52.04, 53.13, 56.05, 74.02, 112.15, 119.42, 120.17, 121.66, 122.43, 126.50, 126.92, 127.05, 127.20, 127.30, 128.46, 130.12, 141.07, 144.58, 145.49, 166.88, 167.23. HRMS (ESI) *m*/*z* calcd for C_32_H_29_N0_5_^+^ (M + H^+^) 508.2118, found 508.2118.

##### Methyl 2,2-dimethyl-6-(2-oxo-2-(pyrazin-2-ylamino)ethoxy)-2H-benzo [h] chromene-5-carboxylate (**6h**)

The spectrum of ^1^H NMR, ^13^C NMR and HRMS of compounds **6h** are shown in Appendix A.

M.p.121–123 °C. ^1^H-NMR (300 MHz, CDCl_3_: δ 1.25 (s, 6H), 3.95 (s, 3H), 4.77(s, 2H), 5.70 (d, 1H, J = 12.0 Hz), 6.49 (d, 1H, J = 9.0 Hz), 7.53–7.56 (m, 3H), 7.92–7.95 (m,1H), 8.23–8.27 (m, 1H), 8.33–8.34 (m, 1H), 8.42–8.43 (m, 1H), 9.26 (s, 1H). ^13^C-NMR (125 MHz, CDCl_3_): δ 27.45, 29.42, 52.43, 73.80, 112.27, 119.44, 120.17, 121.51, 122.57, 126.61, 126.93, 127.08, 127.30, 130.20, 136.83, 140.53, 142.07, 144.33, 145.80, 147.26, 166.76, 166.84. HRMS (ESI) *m*/*z* calcd for C_23_H_21_N_3_0_5_^+^ (M + H^+^) 420.1554, found 420.1554.

##### Methyl 2,2-dimethyl-6-(2-((6-methylpyridin-3-yl)amino)-2-oxoethoxy)-2H-benzo [h] chromene-5-carboxylate (**6i**)

The spectrum of ^1^H NMR, ^13^C NMR and HRMS of compounds **6i** are shown in Appendix A.

M.p.119–121 °C. ^1^H-NMR (300 MHz, CDCl_3_: δ 1.53 (s, 6H), 2.58 (s, 3H), 3.93 (s, 3H), 4.72 (s, 2H), 5.70 (d, 1H, J = 12.0 Hz), 6.43 (d, 1H, J = 12.0 Hz), 7.20 (d, 1H, J = 9.0 Hz), 7.53–7.57 (m, 2H), 7.92–7.95 (m,1H), 8.21–8.26 (m, 2H), 8.69–8.70 (m, 1H), 8.92 (s, 1H). ^13^C-NMR (125 MHz, CDCl_3_): δ 27.60, 29.60, 52.59, 74.30, 112.39, 119.64, 120.04, 121.80, 122.55, 123.08, 127.31, 127.47, 127.66, 130.38, 130.81, 131.52, 140.45, 144.98, 154.41, 166.80, 167.21. HRMS (ESI) *m*/*z* calcd for C_25_H_24_N_2_0_5_^+^ (M + H^+^) 433.1757, found 433.1758.

##### Methyl 2,2-dimethyl-6-(2-oxo-2-(quinolin-8-ylamino)ethoxy)-2H-benzo [h] chromene-5-carboxylate (**6j**)

The spectrum of ^1^H NMR, ^13^C NMR and HRMS of compounds **6j** are shown in Appendix A.

M.p.113–115 °C. ^1^H-NMR (300 MHz, CDCl_3_: δ 1.54 (s, 6H), 3.85 (s, 3H), 4.86 (s, 2H), 5.71 (d, 1H, J = 9.0 Hz), 6.49 (d, 1H, J = 12.0 Hz), 7.45–7.55 (m, 5H), 8.12–8.27 (m, 3H), 8.83–8.93 (m, 2H), 11.21 (s, 1H). ^13^C-NMR (125 MHz, CDCl_3_): δ 28.02, 52.95, 74.65, 75.04, 112.95, 114.09, 117.18, 120.11, 120.96, 122.08, 122.52, 122.62, 122.98, 127.18, 127.52, 127.60, 127.70, 127.90, 128.37, 130.75, 134.30, 136.48, 139.20, 145.46, 146.14, 149.01, 167.34, 167.55. HRMS (ESI) *m*/*z* calcd for C_28_H_24_N_2_0_5_^+^ (M + H^+^) 469.1757, found 469.1758.

##### Methyl 2,2-dimethyl-6-(2-oxo-2-(quinolin-2-ylamino)ethoxy)-2H-benzo [h] chromene-5-carboxylate (**6k**)

The spectrum of ^1^H NMR, ^13^C NMR and HRMS of compounds **6k** are shown in Appendix A.

M.p.116–118 °C. ^1^H-NMR (300 MHz, CDCl_3_: δ 1.54 (s, 6H), 3.39 (s, 3H), 4.73 (s, 2H), 5.73 (s, 1H), 6.45 (d, 1H, J = 9.0 Hz), 7.52–7.59 (m, 6H), 8.22–8.29 (m, 4H), 8.56 (d, 1H, J = 9.0 Hz), 9.51 (s, 1H). ^13^C-NMR (125 MHz, CDCl_3_): δ 27.88, 29.84, 53.56, 72.45, 74.36, 119.92, 120.93, 122.13, 122.57, 122.66, 122.91, 123.58, 125.70, 126.89, 127.01, 127.21, 127.33, 127.44, 127.63, 127.76, 128.55, 130.42, 130.55, 145.88, 150.23, 167.27, 167.76, 169.51. HRMS (ESI) *m*/*z* calcd for C_28_H_24_N_2_0_5_^+^ (M + H^+^) 469.1754, found 469.1758.

#### 3.1.8. General Procedure for the Preparation of Target Compounds **8a–c**

To a mixture of Mollugin **2** in DMF was added K_2_CO_3_ and N-alphatic or N-aromatic heterocycle chroroacetamides **5a–k** in turn. The reaction was stirred for 17 h under refluxing temperatures. Then the solution was diluted with water and extracted 3 times by ethyl acetate to give the crude product. The crude compound was purified by column chromatography (petroleum ether: ethyl acetate = 15:1) to obtain the desired products **8a–c**. The structural formula and yield of compound **8a–c** are shown in Table 8 and further characterized by the physical and spectroscopic data shown below.

##### Methyl 6-(2-bromoethoxy)-2,2-dimethyl-2H-benzo [h] chromene-5-carboxylate (**8a**)

The spectrum of ^1^H NMR, ^13^C NMR and HRMS of compounds **8a** are shown in Appendix A.

M.p.110–112 °C. ^1^H-NMR (300 MHz, CDCl_3_: δ 1.51 (s, 6H), 3.69–3.73 (m, 2H), 3.99 (s, 3H), 4.36–4.40 (m, 2H), 5.67 (d, 1H, J = 12.0 Hz), 6.39 (d, 1H, J = 9.0 Hz), 7.50–7.53 (m, 2H), 8.10–8.13 (m,1H), 8.19–8.22 (m, 1H). ^13^C-NMR (125 MHz, CDCl_3_): δ 27.43, 29.86, 52.24, 74.77, 77.16, 112.03, 119.45, 120.59, 122.20, 126.40, 126.67, 126.73, 127.51, 130.03, 145.01, 145.14, 167.32. HRMS (ESI) *m*/*z* calcd for C_19_H_19_Br0_4_^+^ (M + H^+^) 391.0539, found 391.0539.

##### Methyl 2,2-dimethyl-6-(2-(phenylamino)ethoxy)-2H-benzo[h]chromene-5-carboxylate (**8b**)

The spectrum of ^1^H NMR, ^13^C NMR and HRMS of compounds **8b** are shown in Appendix A.

M.p.135–137 °C. ^1^H-NMR (300 MHz, CDCl_3_: δ 1.52 (s, 6H), 3.49–3.58 (m, 2H), 3.92 (s, 3H), 4.26–4.29 (m, 2H), 5.67 (d, 1H, J = 12.0 Hz), 6.39 (d, 1H, J = 12.0 Hz), 6.73–6.74 (m, 3H), 7.19–7.22 (m, 2H), 7.42–7.52 (m,2H), 7.98–8.00 (m, 1H), 8.19 (d, 1H, J = 9.0 Hz). ^13^C-NMR (125 MHz, CDCl_3_): δ 27.58, 29.59, 43.95, 52.33, 74.13, 76.40, 112.27, 113.04, 117.59, 119.71, 120.61, 122.33, 122.40, 126.63, 126.79, 127.72, 129.21, 130.16, 144.99, 145.91, 147.96, 167.66. HRMS (ESI) *m*/*z* calcd for C_25_H_25_N0_4_^+^ (M + H^+^) 404.1857, found 404.1856.

##### Methyl6-(2-((4-fluorophenyl)amino)ethoxy)-2,2-dimethyl-2H-benzo [h] chromene-5-carboxylate (**8c**)

The spectrum of ^1^H NMR, ^13^C NMR and HRMS of compounds **8c** are shown in Appendix A.

M.p.128–130 °C. ^1^H-NMR (300 MHz, CDCl_3_: δ 1.51 (s, 6H), 3.97 (s, 3H), 4,31–4.33 (m, 2H), 4.50–4.53 (m, 2H), 5.68 (d, 1H, J = 9.0 Hz), 6.42 (d, 1H, J = 12.0Hz), 6.98–7.04 (m, 2H), 7.35–7.39 (m, 2H), 7.48–7.54 (m,2H), 8.02 (d, 1H, J = 9.0 Hz), 8.19–8.22 (d, 1H, J = 9.0 Hz). ^13^C-NMR (125 MHz, CDCl_3_): δ 27.93, 44.98, 52.67, 74.41, 75.28, 112.62, 114.19, 114.25, 115.88, 116.06, 120.04, 120.97, 121.27, 122.61, 122.80, 123.22, 127.00, 127.17, 128.03, 128.37, 130.54, 144.70, 145.39, 146.21, 146.43, 167.68. HRMS (ESI) *m*/*z* calcd for C_25_H_24_FN0_4_^+^ (M + H^+^) 422.1749, found 422.1762.

### 3.2. Biological Evaluation

#### 3.2.1. Cell Culture 

HeLa cells were acquired from the American Type Culture Collection (Manassas, VA, USA) and maintained in Dulbecco’s modified Eagle’s medium (DMEM) supplemented with 10% fetal bovine serum (FBS; Gibco, Grand Island, NY, USA) and 1% penicillin/streptomycin (Invitrogen, Carlsbad, CA, USA) at 37◦C in a humidified incubator containing 5% CO_2_ atmosphere. TNF-α was obtained from R&D Systems (Minneapolis, MN, USA). 

#### 3.2.2. Antibodies and Reagents

Antibody for NF-κB was obtained from BD Biosciences (San Diego, CA, USA). Antibodies for Topo-I were purchased from Santa Cruz Biotechnology (Santa Cruz, CA, USA). Antibody for Alexa flour^®^ 488 goat anti-mouse lgG (H + L) was obtained from (Invitrogen, Carlsbad, CA, USA). MG132, cycloheximide (CHX), propidium iodide (PI), Dimethyl sulfoxide (DMSO), and MTT [3-(4,5-dimethylthiazol-2-yl)-2,5-diphenyl- tetrazolium bromide] were obtained from Sigma Chemical Co (St. Louis, MO, USA).

#### 3.2.3. Plasmids, Transfections, and Luciferase Reporter Assay

A NF-κB-Luc plasmid for NF-κB luciferase reporter assay was obtained from Strategene (La Jolla, CA, USA). Transfections were performed as previously described. NF-κB-dependent luciferase activity was measured using the Dual-luciferase reporter assay system. Briefly, HeLa cells (1 × 10^5^ cells/well) were seeded in a 96-well plate for 24 h. The cells were then transfected with plasmids for each well and then incubated for a transfection period of 24 h. After that, the cell culture medium was removed and replaced with fresh medium containing various concentrations of baicalein for 6 h, followed by treatment with 10 ng/mL of TNF- α for 6 h. Luciferase activity was determined in MicroLumat plus luminometer (EG&G Berthold, Bad Wildbad, Germany) by injecting 100 µL of assay buffer containing luciferin and measuring light emission for 10 s. Co-transfection with pRL-CMV (Promega, Madison, WI, USA), which expresses Renilla luciferase, was performed to enable normalization of data for transfection efficiency [20,21].

#### 3.2.4. Measurement of Cell Viability by MTT Assay

HeLa cells were inoculated and cultured in 96-well plates. After the cells adhered to the wall, the prepared target compounds with different concentrations were added to all plates for 24 h and then cultured in 5 mg/mL MTT solution for 4 h. The culture solution in the holes was sucked out, and DMSO reagent was added. After the crystallization was fully dissolved, the absorbance at 570 nm was measured by Multiskan GO.

#### 3.2.5. Western Blot Assay

The method used for Western blot analysis has been described previously. HeLa cells were seeded into 24-well plates at 1 × 10^4^ cells/well for 24 h. Then, cells were treated with **6d** (80 µM) for 12 h (cells treated with DMSO and 10 ng/mL TNF-α alone were used as negative and positive control, respectively), followed by treatment with 10 ng/mL TNF-α for 30 min. After treatment, cells were rinsed once in PBS, followed by fixation in fresh 4% paraformaldehyde for 30 min at room temperature. Cells were permeabilized with 0.2% Triton X-100 in PBS at room temperature. Next, the cells were incubated with PBS containing 5% BSA for 30 min, and then incubated with the primary antibody against NF-κB p65 at 4 °C overnight. On the second day, the cells were incubated with Alexa flour^®^ 488 goat anti-rabbit lgG (H + L) for 30 min at room temperature, followed by DAPI staining for 30 min before observation. The staining was examined using the Olympus IX83 inverted fluorescence microscope (Olympus Corporation, Tokyo, Japan) at 40× magnification (scale bar 20 µm). The p65 protein showed color in green, and the nuclei showed in blue. The merged images were conducted using Image J software (Wayne Rasband National Institutes of Health, Bethesda, MD, USA) to show co-localization (cyan fluorescence) [22].

### 3.3. Animals

All experimental procedures were conducted in conformity with institutional guidelines for the care and use of laboratory animals at Yanbian University, Jilin, China, and conformed to the National Institutes of Health Guide for Care and Use of Laboratory Animals (Number of license SCXK 2011–0007). All mice were kept in a central animal care facility with free access to water and rodent food during the experiment. 

#### Evaluation of Anti-Inflammatory Activity in Vivo

In vivo inhibition test of xylene induced ear edema in mice was performed to evaluate its anti-inflammatory activity.

In the initial screening, mice (male) were randomly divided into experimental group, positive control drug ibuprofen and Mesalazine group, and blank group. All target compounds were dissolved in dimethyl sulfoxide, and corresponding drugs were intraperitoneally injected into experimental group, positive control group and blank group, respectively. Mice were treated with 100 mg/kg, 0.1 mL/20 g body weight, and control mice received only DMSO (0.1 mL/20 g body weight). After 30 min, 20 μL of xy-lene was evenly applied on both sides of the right ear of the mice. After 30 min of xy-lene was applied, the mice were sacrificed by cutting off their necks. Then the left and right ears of the mice were drilled with a hole punch with a diameter of 7 mm, respec-tively, to obtain two round ear pieces. The degree of swelling (right ear mass–left ear mass) was calculated for each sample, and the swelling inhibition rate of each drug was calculated using the following formula. The inhibition rate of swelling degree (%) = (mean ear swelling degree of mice in blank group and mean ear swelling degree of mice in target compound group)/mean ear swelling degree of mice in blank group × 100%. The data were processed by GraphPadPrism 5 and expressed as mean data ± SEM. One-way ANOVA method was used for statistical analysis of relevant data. A value of *p* < 0.05 in the chart indicates significant and statistically significant data dif-ference [3].

In the deep screening, mice were divided into target compound group, positive control group (ibuprofen), and blank group (8 male mice in each group). Target com-pound and ibuprofen were dissolved in sodium carboxymethyl cellulose aqueous solu-tion (sodium carboxymethyl cellulose to water, mass ratio 1:1000); the blank group was sodium carboxymethyl cellulose aqueous solution (sodium carboxymethyl cellu-lose to water, mass ratio 1:1000). The anti-inflammatory activities of each compound in the target compound group, positive control group, and blank group were measured at 1, 2, 3, 4, 6, 9, 12 and 24 h after taking the drug. The experimental operation and calculation method are the same as the initial screening part.

### 3.4. Prediction of ADMET Properties

ADMET properties of target compounds **4f** and **6d** as drug lead compound were predicted using ADMET descriptors in Discovery Studio 2017 (Accelrys, San Diego, CA, USA) [20]. It is a quick, easy, and accurate method for prediction of absorption, distribution, metabolism, elimination, and toxicity (ADMET) properties. In this work, for the aforementioned compounds, human intestinal absorption level, aqueous solubility (log (SW), blood brain barrier (BBB), penetration level (AlogP98), human cytochrome P4502D6 (CYP2D6) inhibitory ability, hepatotoxicity possibility, and plasma protein binding (PPB) level were measured.

## 4. Conclusions

In conclusion, several series of mollugin derivatives were designed, synthesized, and evaluated for NF-κB inhibitory activity and toxicity. Of the compounds tested, **6d** showed the most promising biological profile, exhibiting NF-κB inhibitory activity (IC_50_ value) of 3.81 µM, and did not show any significant cytotoxicity. In addition, Western blotting indicated that compound **6d** inhibited TNF-α-induced expression of p65 at a concentration of 100 µM. The results of in vivo anti-inflammatory tests showed that most of the compounds displayed more potent inhibitory activity than the parent compound, and compound **4f** was the most potent with an inhibition rate of 83.08%, which was significantly more potent than mollugin and the positive control drugs (ibuprofen and mesalazine). The results of an ADMET prediction experiment indicated that compounds **4f** and **6d** displayed good pharmacokinetic and drug-like behaviors. These results provide an initial basis for the development of **4f** and **6d** as potential anti-inflammatory agents.

## Data Availability

Not applicable.

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
