# Peer review of "Synthesis and Evaluation of NF-κB Inhibitory Activity of Mollugin Derivatives"

_molecules, 2022, doi:10.3390/molecules27227925_

Round 1

Reviewer 1 Report

The authors describe the synthesis and the biological evaluation of a set of Mollugin derivatives obtained by structurally modifying the phenolic group at C-6 position of the lead compound.

The manuscript is well-written, and the experimental plan is appropriately set up.

However, before publication, it’s necessary to address the following issues:

1)      Concerning the Luciferase assay, the authors should better specify the composition of the co-transfected plasmids used, in order to make the analysis’s results clearer.

2)      It’s necessary to include in the main text a higher resolution Western Blot imagine because it’s quite hard to appreciate the dose-dependent down modulation of p65. Currently, it is only possible to observe a decreased level of p65 at 100 mM, a concentration, by the way, much higher than that necessary in Luciferase assay to produce the 50% of NF-Kb inhibition.

3)      The in vivo data, in which the xylene-induced ear edema reduction in mice is measured, show some discrepancies that deserve to be better clarified. Indeed, aside from the interesting results found for the disclosed hits 4f and 6d, there are also other compounds (like 8b, 6j and 6h) showing promising results. These latter, likely, have to be ascribed to a different mechanism of action as, for example, compound 8b, displaying an anti-inflammatory activity comparable to that of 6d (76.52 and 76.77, respectively), is completely unable to determine NF-Kb inhibition. Could the authors provide comments and convincing explanations about this topic?

Author Response

Dear Editor and Reviewers,

     Thanks very much for taking your time to review this manuscript. I really appreciate all your comments and suggestions! Please find my itemized responses in below and my revisions in the re-submitted files. We hope this modification can meet your requirements. Looking forward to receiving your reply!

Reviewer 2 Report

The authors made great efforts to Synthesize and evaluate the inhibitory activities of novel Mollugin derivatives, the following suggestions and comments may enhance the manuscript's readability:

-         Check language editing, eg: Line 91, General procedures for….

-         2.1.6.1-8: remove the yield of each compound and collect them in one table or figure for comparison.

-         Line 198, I think it is a new compound, so can take 2.1.6.9 ??

-         Table 1: Remove 4 a-I structures, and collect all structures in a separate table with their structure.

-         Enhance the resolution of figures 2,3, and 4.

 - In supplementary files, add figures number ( Figure S1, …..), apply for all

Author Response

(The authors gave the same response as above.)

Reviewer 3 Report

The authors designed and synthesized 23 mollugin derivatives and evaluated their inhibitory activity against NF-κB transcription. 6d emerged as the most potent compound with potent anti-inflammatory activity. The synthesis is straightforward and interesting. The article is well drafted with minor spelling and grammar mistakes. In vivo should be 'In vivo' in the entire manuscript.

Moreover, 13C NMR of  two compounds should be revised (6a and 6k) with more scans. In addition, it is not clear if the  in vitro cytotoxicty against normal cells was evaluated ? If yes, which normal cell lines were used? If not evaluated, then the in vitro cytotoxicty against normal cells for the most active compounds should be performed by the authors which can give a clear picture about the selectivity index.

Author Response

(The authors gave the same response as above.)

Round 2

Reviewer 2 Report

Many thanks for the authors, as they respond to all suggestions and comments